# Whole Genome Sequencing and Comparative Analysis of the First *Ehrlichia canis* Isolate in China

**DOI:** 10.3390/microorganisms12010125

**Published:** 2024-01-08

**Authors:** Jilei Zhang, Jiawei Wang, Chengming Wang

**Affiliations:** 1College of Veterinary Medicine, Yangzhou University, Yangzhou 225009, China; wangjiawei@cwbio.cn; 2College of Medicine, University of Illinois Chicago, Chicago, IL 60612, USA; 3College of Veterinary Medicine, Auburn University, Auburn, AL 36849, USA

**Keywords:** *Ehrlichia canis*, whole genome sequencing, tick-borne pathogens, dogs

## Abstract

*Ehrlichia canis*, a prominent tick-borne pathogen causing canine monocytic ehrlichiosis (CME), is one of the six recognized *Ehrlichia* species worldwide. Despite its widespread presence in ticks and host dogs in China, comprehensive genomic information about this pathogen remains limited. This study focuses on an in-depth analysis of *E. canis* YZ-1, isolated and cultured from an infected dog in China. The complete genome of *E. canis* YZ-1 was sequenced (1,314,789 bp, 1022 genes, 29% GC content, and 73% coding bases), systematically characterizing its genomic elements and functions. Comparative analysis with representative genomes of *Ehrlichia* species, including *E. canis* strain Jake, *E. chaffeensis*, *Ehrlichia* spp., *E. muris*, *E. ruminantium*, and *E. minasensis*, revealed conserved genes, indicating potential evolutionary connections with *E. ruminantium*. The observed reduction in virulence-associated genes, coupled with a type IV secretion system (T4SS), suggests an intricate balance between pathogenicity and host adaptation. The close relationship with *E. canis* Jake and *E. chaffeensis*, alongside nuanced genomic variations with *E. ruminantium* and *E. mineirensis*, underscores the need to explore emerging strains and advancements in sequencing technologies continuously. This genetic insight opens avenues for innovative medications, studies on probiotic resistance, development of new detection markers, and progress in vaccine development for ehrlichiosis. Further investigations into the functional significance of identified genes and their role in host–pathogen interactions will contribute to a more holistic comprehension of *Ehrlichia*’s biology and its implications for pathogenicity and transmission.

## 1. Introduction

Ehrlichiosis, attributed to a cluster of emerging rickettsial tick-borne pathogens, comprises Gram-negative obligate intracellular bacteria within the genus *Ehrlichia*. The six recognized species within this genus include *E*. *canis*, *E*. *chaffeensis*, *E*. *ewingii*, *E*. *muris*, *E. minasensis*, and *E*. *ruminantium* [1,2,3]. Of particular significance is *E. canis*, the causative agent of canine monocytic ehrlichiosis (CME), also known as canine rickettsiosis, canine typhus, canine hemorrhagic fever, tropical canine pancytopenia, and tracker dog disease. This disease targets platelets, monocytes, and granulocytes, gaining widespread recognition in the 1970s when significant mortalities were observed in German Shepherd dogs during the Vietnam War, despite its initial description in Algeria in 1938 [4,5,6].

While most dogs with CME recover through treatment with doxycycline or rifampicin, alongside appropriate supportive care, chronic cases leading to fatalities are not uncommon [4,7]. Notably, *E. canis* infections have also been reported in humans [8,9].

Despite several decades of study, genetic resources for *E. canis* remain sparse, with only one complete genome of *E. canis* strain Jake, initially isolated in North America and published in 2005 [10,11], and some from Australia [12]. The challenges of in vitro cultivation of *E. canis* and the complexities of obtaining genomes from their intracellular locations contribute to this scarcity [4,13]. To address this, we include representative genomes of all six *Ehrlichia* species [14]. Typically, *Ehrlichia* genomes range from 1.2 to 1.5 Mbp, containing approximately 1000 genes, making them relatively small compared with most bacteria [14,15].

In light of these challenges, we embarked on a genomic analysis of *E. canis* isolated and cultured in the laboratory from an infected dog during an ehrlichiosis epidemiological investigation in China. Our primary objective was to acquire genomic data specific to the Chinese strain, enabling comparative analyses with other publicly available *Ehrlichia* genomes. This genomic dataset and analysis aim to enhance our understanding of the relatedness and evolution of *E. canis* in China, providing valuable insights into potential molecular markers for detection, treatment, and vaccination against CME.

## 2. Materials and Methods

### 2.1. Preparation of E. canis for Sequencing

The canine monocytic cell line DH82 (ATCC CRL-10389, ATCC, Manassas, VA, USA) was cultivated in Minimum Essential Medium Eagle (MEM) (Sigma-Aldrich, Saint Louis, MO, USA) medium, supplemented with 10% fetal bovine serum (Gibco, Billings, MT, USA) as described before [4]. The *E. canis* strain YZ-1 used in this study was isolated from an adult female beagle dog (*Canis lupus familiaris*) living in a commercial canine facility in Taizhou, Jiangsu, China. This strain was maintained in our laboratory by continuous passage for more than 20 passages in DH82 cells [15]. For propagation, the *E. canis* was inoculated onto a monolayer of DH82 cells and incubated for six days at 37 °C under 5% CO_2_. Subsequently, the culture medium and cells were harvested using a 23 cm cell scraper (Thermo Fisher Scientific, Waltham, MA, USA) and preserved at −80 °C until nucleic acid extraction.

### 2.2. Nucleic Acid Purification

The suspension of pathogens and cells was centrifuged at 500× rpm, 25 °C for 5 min to remove the cell debris. Then, the supernatant was transferred to a new tube, followed by centrifugation at 14,000× rpm, 25 °C for 15 min. The supernatant was removed, and the pellets were resuspended with a sucrose–phosphate–glutamate (1:1:1) solution. The DNAs were extracted from the suspension with a High Pure PCR Template Preparation Kit per the manufacturer’s description [3,16,17].

### 2.3. Whole Genome Sequencing and Assembly

Total DNA obtained for WGS underwent quality control, involving the electrophoretic separation of 1 µL of DNA on an agarose gel and quantification using the Qubit system. The genome of *E. canis* YZ-1 was sequenced utilizing Single-Molecule, Real-Time (SMRT) technology on the 3rd generation PacBio platform, conducted at Beijing Novogene Bioinformatics Technology Co., Ltd. (Beijing, China). The SMRT Analysis 2.3.0 software was employed to filter low-quality reads and assemble them into a single, gap-free contig using the filtered reads. Subsequently, the genome underwent de novo assembly through the SMRT portal software, guided by the valid sequencing data. The automatically annotated *E. canis* YZ-1 genome draft was generated using the NCBI Prokaryotic Genomes Annotation Pipeline (NCBI_PGAP). The complete genome sequence was deposited in the NCBI database with the GenBank accession number CP025749 [15].

### 2.4. Ehrlichia canis YZ-1 Genome Analysis

#### 2.4.1. Genomic Components Analysis

Following the acquisition of the complete genome of *E. canis* YZ-1, various genomic components, including coding genes, repeats, noncoding RNA (ncRNA), and genomic islands (GIs), were predicted utilizing specialized software. Coding genes were identified and analyzed using GeneMark, a widely employed tool for gene prediction in bacteria, archaea, and metagenomics [18]. The analysis of repeats, encompassing tandem repeats (TR) and interspersed repeats (IR), was conducted through the Tandem Repeats Finder (TRF) and RepeatMasker software 4.1.6, respectively [19,20]. The prediction of ncRNA, crucial in life activities without transcription, involved rRNA database blasting or RNAmmer software for rRNA, tRNAscan software for tRNA location and structure, and Rfam software for sRNA [21,22,23]. Indicative of horizontal origins with potential involvement in symbiosis or pathogenesis, genomic islands were predicted using IslandPath-DIMOB software.

#### 2.4.2. Genomic Function Annotation

The annotation of genes in *E. canis* YZ-1 was carried out using multiple databases, including Gene Ontology (GO), the Kyoto Encyclopedia of Genes and Genomes (KEGG), the Cluster of Orthologous Groups of Proteins (COG), the Non-Redundant Protein Database (NR), and Swiss-Prot. The GO knowledgebase [24] encompasses three main categories—cellular component, molecular function, and biological process—providing comprehensive bioinformatics information on gene functions. KEGG [25,26], a comprehensive database, facilitated the systematic analysis of cellular metabolic pathways and gene expression functions in organisms. COG, NR, and Swiss-Prot [27] are protein-specific annotation databases focusing on different protein functions, such as nonredundant protein information in NR.

Additionally, pathogenesis and antibiotic resistance genes were analyzed using the *E. canis* YZ-1 genome. The PHI (Pathogen–Host Interactions) database [28] annotated interactions between *E. canis* and hosts by blasting target amino acid sequences with reference sequences in PHI. SignalP software 5.0 [29] was employed for the recognition of secretion proteins, while EffectiveT3 software was utilized for the type recognition of the secretion system [30]. Simultaneously, the VFDB (Virulence Factors of Pathogenic Bacteria) database [31] was used to identify virulence factors in *E. canis* YZ-1, and the ARDB (Antibiotic Resistance Genes Database) was consulted for antibiotic resistance analysis [32].

### 2.5. Macroscopic Comparative Genomic and Phylogenetic Analysis

The genome content of *E. canis* YZ-1 (CP025749) underwent a thorough comparison analysis against the reference genome of *E. canis* strain Jake (NC_007354) and other publicly available ehrlichial species, including *E. chaffeensis* (CP000236), *Ehrlichia* spp. (NZ_CP007474), *E. muris* (CP006917), *E. ruminantium* (CR767821), and *E. minasensis* (NZ_CDGH01000070). Pairwise genomic comparisons were conducted using the Artemis Comparison Tool (ACT) [33] and Geneious 9 [34] with alignments produced by progressive Mauve [35] and MAFFT [36]. Whole genome comparisons were performed with Blastn and tBlastx algorithms using BRIG [37] and Easyfig [38]. The graphical representation of the *E. canis* YZ-1 genome and its components was achieved using DNA-Plotter [39], providing a visual overview of the genomic landscape and facilitating a comprehensive understanding of genomic variations and relationships among the studied *Ehrlichia* species.

## 3. Results

### 3.1. E. canis YZ-1 Genome Assemblies and Characteristics

Utilizing next-generation sequencing technologies, we successfully sequenced, assembled, and characterized the whole genome of *E. canis* strain YZ-1 [4]. This enabled a comprehensive pairwise comparison with other *E. canis* and *Ehrlichia* strains available in publicly accessible databases. The whole genome of *E. canis* YZ-1 spans 1,314,789 bp, harboring 1022 genes, exhibiting an overall GC content of 29%, and comprising 73% coding bases (Table 1, Figure 1 and Appendix A). In our comparative analysis, 900 out of 1022 genes demonstrated identical sequences to those of *E. canis*, albeit with little similarity to other species, including *E. chaffeensis* (6/1022), *E. muris* (2/1022), *Wolbachia* WPWAU_1221 (1/1022), *Ehrlichia* spp. (1/1022), and *Bacillus* sp. (1/1022) (Appendix A).

Repetitive sequences, and interspersed and tandem repeats, are vital in genome structure maintenance. For *E. canis* YZ-1, we identified 34 long-terminal repeats with a total length of 2405 bp, 14 transposons with a combined size of 1339 bp, and 105 tandem repeats, constituting approximately 2.6% of the entire genome (Table 2) [40].

We thoroughly analyzed noncoding RNA, specifically rRNA, which is essential for ribosome composition and protein translation. Our analysis identified one 16S rRNA (2738 bp), one 5S rRNA (115 bp), and one 23S rRNA (2791 bp), along with 36 tRNA species totaling 2794 bp (Figure 1). Additionally, a genomic island spanning 44,886 bp was identified at positions 283,341 bp to 328,226 bp in the *E. canis* YZ-1 genome (Figure 1).

DNA methylation, a crucial biological process involving adding methyl groups to DNA molecules, can influence the activity of DNA segments without altering the sequence. In *E. canis* YZ-1, we identified 3730 DNA methylation locations, including 205 (5.5%) N6-methyladenine (6 mA), 391 (10.5%) N4-methylcytosine (4 mC), and an overwhelming majority of unknown methylation patterns (84.0%, 3134/3730) (Appendix A). Notably, 5-methylcytosine (5 mC) was not detected in the analyzed methylation types. (Appendix A) [41,42,43].

### 3.2. Genomic Functional Analysis

The Gene Ontology (GO) project offers structured, controlled vocabularies classifying molecular and cellular biology into three nonoverlapping domains: molecular function, cellular component, and biological process, with further subdomains. In our analysis of 1022 genes from *E. canis* YZ-1, 2959 annotations were obtained through GO analysis (Figure 2A). Notably, a substantial number of genes were associated with cellular processes (450 genes) and metabolic processes (453 genes) within the biological process domain. Similarly, the cellular component domain revealed a predominant presence of genes related to cell (268) and cell part (268) (Figure 2A). In the molecular function domain, the most represented functions were binding (333) and catalytic activity (399) (Figure 2A) [44].

KEGG, a database integrating genomic and higher-order functional information, provides graphical representations of cellular processes. More than half of the genes in *E. canis* YZ-1 (537 out of 1022) were associated with 97 pathways in KEGG. Further analysis of subpathways revealed categories with over 50 genes, including translation-related pathways (74 genes) in genetic information processing, nucleotide metabolism (50 genes), and metabolism of cofactors and vitamins (59 genes) in metabolism (Figure 2B) [45].

The COGs database, based on the orthology concept, classifies proteins from completely sequenced genomes. In the *E. canis* YZ-1 genome, proteins were classified using the COG database, enhancing the genome sequence’s utility for functional and evolutionary analysis. The highest enrichment was observed in genes coding for proteins related to translation, ribosomal structure, and biogenesis (138 genes), followed by energy production and conversion (70 genes), coenzyme transport and metabolism (63 genes), post-translational modification, protein turnover, and chaperones (61 genes), and replication, recombination, and repair (53 genes) (Figure 2C) [46,47].

As a zoonotic pathogen, *E. canis* threatens animals, humans, ecosystems, and regional economies. The PHI database was consulted to understand the dynamic interactions between pathogens and their hosts. In *E. canis* YZ-1, 25 genes were related to reduced virulence, the most prevalent among identified PHI phenotypes (Figure 2D). Additionally, four genes related to increased virulence (hypervirulence) were identified, showing varying similarities to genes from other organisms (Figure 2D, Appendix A).

Moreover, six genes in *E. canis* YZ-1 were associated with virulence genes *CcmC*, *Hsp60*, *ClpC*, *SodB*, *LPS*, and *ClpP*, albeit with low similarity percentages ranging from 40.8% to 58.4%. Analyzing the SignalP server and TMHMM revealed 35 signal peptides and 300 transmembrane helices proteins. Interestingly, 35 of these proteins possessed signal peptides but lacked transmembrane helices. VirB genes related to the type IV secretion system were identified, consisting of two clusters containing virB8/virB9/virB10/virB11/virD4 and virB3/virB4/three large virB6, along with three virB9, virB8, and virB4 located separately (Appendix A). Unfortunately, no gene related to antibiotic resistance was found in *E. canis* YZ-1 [48].

### 3.3. Similarity between E. canis YZ-1 and Other Ehrlichia Species/Strains

#### 3.3.1. Synteny Comparative Genome Analysis

Synteny analysis, employing orthologous genes with similar functions, elucidates homologous gene conservation and order across genomes. In our study, we conducted a comprehensive synteny analysis comparing *E. canis* YZ-1 with various *Ehrlichia* species/strains, including *E. chaffeensis*, *Ehrlichia* spp., *E. muris*, *E. ruminantium*, *E. mineirensis*, and *E. canis* Jake [49] (Table 1, Appendix A).

When aligned with *E*. *chaffeensis* strain Arkansas (CP000236), which was isolated from a patient resident in Arkansas [10,11], there were 76.7% (1,008,696/1,314,789 bp) of *E*. *canis* YZ-1 and 85.9% (1,010,920/1,176,248) of *E*. *chaffeensis* involved in both bidimensional (Appendix A) and parallel (Figure 3) comparative analysis. In all the 337 collinear alignment blocks, there were 309 collinearities with a similarity between 48.8% and 91.1%, nine inversions with a similarity between 70.0% and 84.3%, 12 translocations with a similarity between 47.9% and 81.8%, and six inversions/translocations with a similarity between 64.5% and 87.8% (Figure 3A and Appendix A).

When aligned with *Ehrlichia* spp. HS strain (*Ehrlichia japonica*) (NZ_CP007474) [50], there were 77.48% (1,018,665/1,314,789 bp) of *E*. *canis* YZ-1 and 89.0% (1,022,729/1,148,904) of *Ehrlichia* spp. involved in this analysis. In all the 294 collinear alignment blocks, there were 183 collinearities with a similarity between 45.6% and 90.8%, nine inversions with a similarity between 75.0% and 86.8%, 12 translocations with a similarity between 59.0% and 82.6%, and 92 inversions/translocations with a similarity between 54.2% and 88.1% (Figure 3B and Appendix A).

When aligned with *E*. *muris* strain AS145 (CP006917), which was isolated from a wild mouse in Japan [51], there were 77.1% (1,013,862/1,314,789 bp) of *E*. *canis* YZ-1 and 85.0% (1,017,665/1,196,717) of *E*. *muris* involved in this analysis. In all the 317 collinear alignment blocks, there were 201 collinearities with a similarity between 46.2% and 93.1%, 15 inversions with a similarity between 67.1% and 84.8%, ten translocations with a similarity between 51.4% and 88.9%, and 91 inversions/translocations with a similarity between 54.8% and 89.7% (Figure 3C and Appendix A).

When aligned with the *E*. *ruminantium* Welgevonden-type strain (CP006917), which was isolated from cattle in Welgevonden, South Africa [52], there were 68.1% (895,013/1,314,789 bp) of *E*. *canis* YZ-1 and 59.3% (898,925/1,516,355) of *E*. *ruminantium* involved in this analysis. In all the 434 collinear alignment blocks, there were 241 collinearities with a similarity between 50.0% and 86.7%, 17 inversions with a similarity between 62.3% and 81.0%, 28 translocations with a similarity between 45.5% and 96.4%, and 148 inversions/translocations with a similarity between 35.0% and 91.4% (Figure 3D and Appendix A).

*E*. *mineirensis* is a new *Ehrlichia* species isolated from *Rhipicephalus* (*Boophilus*) *microplus* from Minas Gerais, Brazil, and has been molecularly characterized recently [14,53]. When aligned with this strain (NZ_CDGH01000070), there were 95.9% (1,260,937/1,314,789 bp) of *E*. *canis* YZ-1 and 89.1% (1,261,326/1,414,910) of *E*. *mineirensis* involved in this analysis. In all the 335 collinear alignment blocks, there were 80 collinearities with a similarity between 62.8% and 92.9%, 13 inversions with a similarity between 46.6% and 90.6%, 114 translocations with a similarity between 56.7% and 96.1%, and 128 inversions/translocations with a similarity between 39.5% and 93.3% (Figure 3E and Appendix A).

When aligned with *E*. *canis* strain Jake (NC_007354) [54], there were 99.3% (1,305,712/1,314,789 bp) of *E*. *canis* YZ-1 and 99.3% (1,305,822/1,315,030) of *E*. *canis* Jake involved in this analysis. In all the 38 collinear alignment blocks, there were 17 collinearities with a similarity between 91.0% and 100%, five inversions with a similarity between 79.3% and 99.96%, zero translocation, and 16 inversions/translocations with a similarity between 77.7% and 100%, which again proved that our strain is *Ehrlichia canis*, not another *Ehrlichia* species (Figure 3F and Appendix A).

#### 3.3.2. Insertion–Deletion Mutations Analysis

Indels, or insertion–deletion mutations, are the insertions and/or deletions of nucleotides into and/or out of genomic DNA with less than 1kb length. They are supremely critical in clinical next-generation sequencing, due to their driving mechanism underlying many constitutional and oncological diseases [55]. We identified these mutations by comparing our *E*. *canis* strain YZ-1 to other *Ehrlichia* species or strains. The indels mainly happened inside the CDS (coding sequencings), with a higher number in *E*. *chaffeensis* (13 insertions; 20 deletions) and *E*. *canis* (17 insertions; 16 deletions) compared with the other species/strains (Appendix A). However, these insertion/deletion mutations were extremely short (less than 1 bp) and mostly frameshift mutations or without impacts on the open reading frame (Appendix A).

#### 3.3.3. Structural Variation (SV) Analysis

Structural variations (SVs) are generally defined as a region of DNA approximately 50 bp long or longer and can include inversions and balanced translocations or genomic imbalances, such as insertions and deletions. These SVs often overlap with segmental duplications, DNA regions present more than once, or copies of which more than 90% of genes are identical [56,57,58]. Here, we performed the SVs analysis on our *E*. *canis* YZ-1 and the reference *Ehrlichia* species one-to-one to identify the regions and types of SVs (Figure 4 and Appendix A). Compared with *E*. *canis* YZ-1, *E*. *chaffeensis* (334), *Ehrlichia* spp. (393), *E*. *muris* (402), and *E*. *ruminantium* (525) have the most SVs, mainly complex iIndels, which are not comparable due to the significant mutations that occurred in that region (Figure 4A–D and Appendix A). Meanwhile, *E*. *mineirensis* has 147 SVs, mainly insertions and inversions (Figure 4E and Appendix A). However, the *E*. *canis* Jake strain only has 42 SVs, mainly insertions and deletions (Figure 4F and Appendix A).

## 4. Discussion

In this study, we conducted next-generation sequencing and comparative genomic analysis on *E. canis* YZ-1, an isolate which was cultured from an infected dog from a commercial canine farm. Our investigation aimed to unravel pathogenesis at the gene level and establish genetic relatedness with other *Ehrlichia* species/strains. Notably, our findings contribute to a deeper understanding of this zoonotic pathogen’s evolutionary dynamics and functional aspects.

Our genomic analysis confirmed *E. canis* YZ-1’s identity, revealing substantial genomic congruence with the *E. canis* Jack strain and a close relationship with *E. chaffeensis* and *E. ruminantium*. Intriguingly, we observed a notable reduction in virulence-associated genes, suggesting an adaptive strategy favoring prolonged survival within the host, aligning with observations of asymptomatic infections in our experimental dog model [4]. The presence of genes such as *ClpP* and *ClpX*, involved in protease activity and ATP-dependent processes, underscores the pathogen’s intricate regulation of virulence [59].

Pathogen–host interactions revealed a plethora of genes with low similarity to those in the database. While the mechanisms remain elusive, these findings align with our previous observations of infected dogs recovering without treatment, indicating potential adaptations for a prolonged host survival period [4]. Identifying a type IV secretion system (T4SS) suggests a role in translocating bacterial effectors into host cells, further highlighting the intricate nature of *E. canis* YZ-1’s interactions with its host. Similar to the *E. canis* reported before, we found two clusters of Vir homologous proteins in *E*. *canis* YZ-1. One contains virB8/virB9/virB10/virB11/virD4, and one includes virB3/virB4/virB6/virB6/virB6 [54]. But we also found virB9, virB8, and virB6 located between and downstream of these two clusters, which was also reported in the *Ehrlichia* spp. HF recently [60].

Comparative genetic analysis showcased *E. canis* YZ-1’s almost complete synteny with the *E. canis* Jake strain, affirming its species identity, which was reported to be almost identical to *Ehrlichia ruminantium* (Welgevonden and Gardel strains) before [54]. However, *E*. *canis* YZ-1 has the lowest similarity with the *E*. *ruminantium* Welgevonden strain, with only 68.1% of *E*. *canis* YZ-1 and 59.3% of *E*. *ruminantium* involved in the analysis. Instead, *E*. *canis* YZ-1 is closer to *E*. *mineirensis* (95.9% of *E*. *canis* YZ-1 and 89.1% of *E*. *mineirensis*), a new *Ehrlichia* strain reported recently [14,53], possibly influenced by host specificity and environmental factors.

One limitation of our analysis was not to include the first *E. canis* strain in Australia [12]. The genomic comparison in this study was conducted before we had access to the genome sequences of the Australian strain. Notably, Neave et al. conducted a comprehensive genomic analysis of the isolates from Australia, the Jake strain, and the YZ-1, and concluded that the Australian *E. canis* genomes were highly conserved, and the Australian genomes were most similar to *E. canis* YZ-1 from China [12].

Interestingly, our analysis revealed a closer relationship between *E. canis* YZ-1 and *E. canis* strains from Australia than those from other regions. Phylogenetic analysis using different genes demonstrated varying relationships, highlighting the importance of comprehensive analyses involving multiple genetic markers (Appendix A). Notably, the inclusion of the variable *trp36* gene pointed to the Australian strains belonging to the Taiwanese genotype, emphasizing the need for a nuanced understanding of strain diversity [61].

This close relationship between *E. canis* YZ-1 and *E. ruminantium* may also be supported by the finding that multiple membrane transporter genes in our *E. canis* YZ-1 are originally from *E. ruminantium* when the analysis was performed based on the Transporter Classification Database, which is a database with a comprehensive IUBMB approved classification system for membrane transport proteins known as the transporter classification (TC) system. There are 62 membrane transport proteins that were identified from *E. canis* YZ-1, and 11 of them (accession #: Q5HB83; Q9R425) are from *E. ruminantium* (Appendix A). Interestingly, 10 out of the 11 proteins were major antigenic proteins (Q9R425). However, the reason behind this is still unclear. One of the possibilities is that both pathogens have infected the same host (ticks or animals) and shared their genomic information. For instance, an *Ehrlichia* species close to *E. canis* and *E. ruminantium* has been collected from the same host camel [62]. Moreover, there are 11 proteins in *E. canis* YZ-1 related to ubiquinone (ubiquitous or coenzyme Q in humans and animals, respectively) biosynthesis. These proteins or coding genes were also reported as originally coming from *E. ruminantium* and *E. chaffeensis* [58]. Further studies on the divergence or characterization of these genes and proteins should be performed in establishing therapeutic interventions in the ongoing battle against tick-borne pathogens. Moreover, a more in-depth exploration of evolutionary history and potential reasons behind these observations requires further studies.

## 5. Conclusions

The comprehensive genomic sequencing of *E. canis* YZ-1 has furnished essential resources for an in-depth exploration of this pathogen, offering valuable insights into host-pathogen interactions, potential evolutionary trajectories, and genetic distinctions from globally prevalent *E. canis* strains. The abundance of genes associated with virulence reduction, potentially aiding evasion of the host immune system, coupled with its close relationship to *E. ruminantium*, presents intriguing avenues for future research and analysis. Unraveling the mechanisms underlying these genetic attributes is imperative for a nuanced understanding of *E. canis* YZ-1’s adaptive strategies. The proteins implicated in these processes and those mediating pathogen-host interactions emerge as promising candidates for developing vaccines and therapeutic interventions against ehrlichiosis. Further studies are warranted to decipher these genes’ functional significance and role in the intricate dynamics of host-pathogen relationships. By unraveling the molecular intricacies of *E. canis* YZ-1, this study lays the groundwork for advancing our comprehension of ehrlichial pathogenesis and enhancing disease prevention and control strategies. Continued investigations into the multifaceted aspects of *E. canis* YZ-1’s biology will contribute to our understanding of ehrlichiosis and offer potential avenues for targeted therapeutic interventions in the ongoing battle against tick-borne pathogens.

## Figures and Tables

**Figure 1 microorganisms-12-00125-f001:**
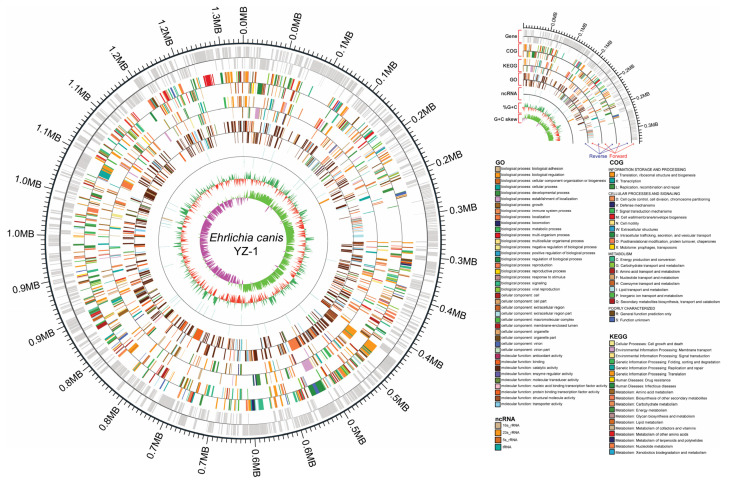
The complete genome of *Ehrlichia canis* YZ-1 from Yangzhou, China. As indicated in the right panel of the figure, there are six tracks of the *E. canis* YZ-1 genome from inside to outside: gene coding sequences in 1000 bp windows, COG annotation, KEGG annotation, GO annotation, noncoding (nc) RNA location, percentage of nucleotides G + C in the genome, and G + C skew values. Each piece of genomic information was shown in forward (upside) and reverse (downside) directions. Moreover, the color map of the GO, COG, KEGG, and ncRNA was included in the figure.

**Figure 2 microorganisms-12-00125-f002:**
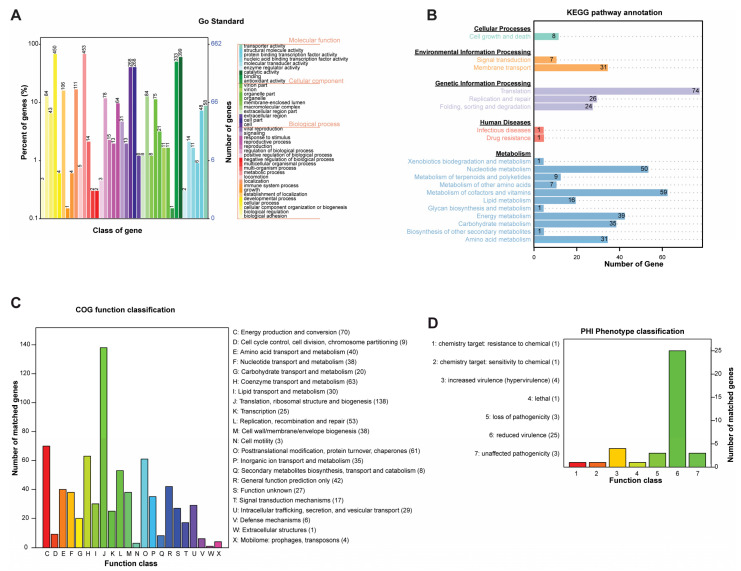
Genomic functional analysis of *Ehrlichia canis* YZ-1. (**A**) The functional analysis of genes, both number (*Y*-axis on the right) and percentage of total genes (*Y*-axis on the left), from *E. canis* YZ-1 was performed using the GO database (*X*-axis), which is established based on the biological process. (**B**) The number of genes (*X*-axis) from the *E. canis* YZ-1 in each metabolic pathway (*Y*-axis) was analyzed using the KEGG database. (**C**) The number of genes (*Y*-axis) of *E. canis* YZ-1 in biological functions (*X*-axis) was shown based on the COG database. (**D**) The number of genes from *E. canis* YZ-1 in each Pathogen–Host Interactions (PHI) category was shown here.

**Figure 3 microorganisms-12-00125-f003:**
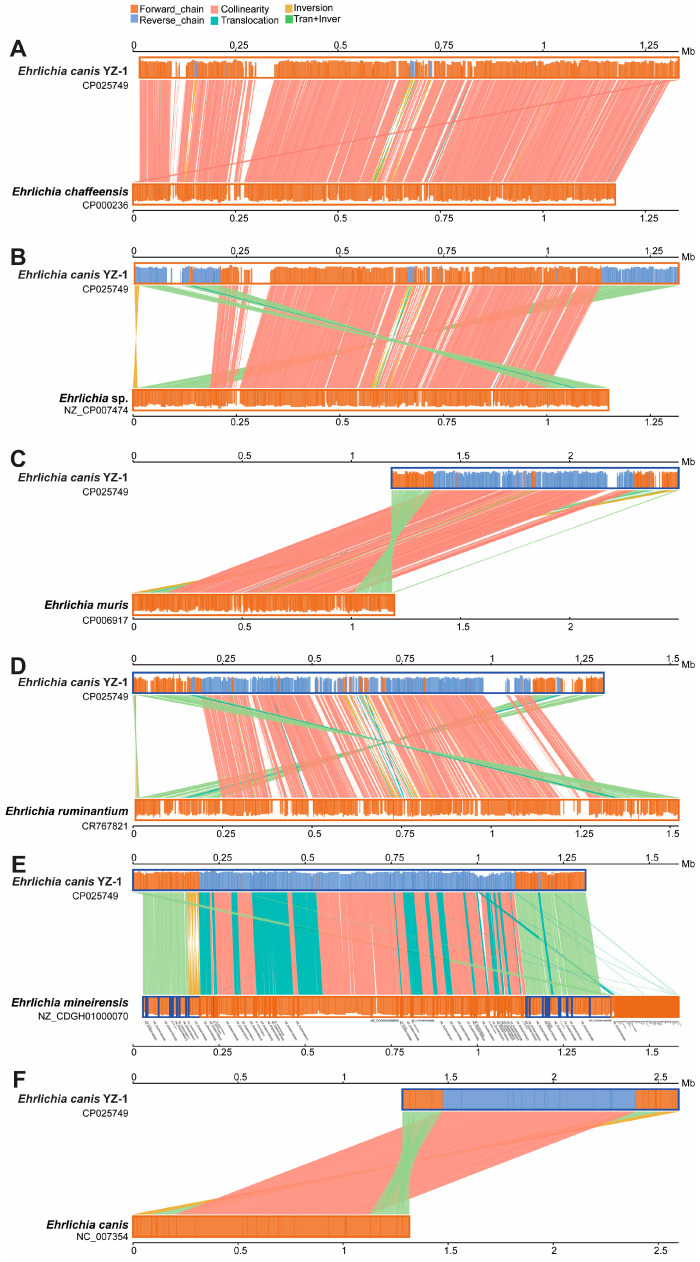
Parallel synteny analysis of *Ehrlichia canis* YZ-1 with the *Ehrlichia* reference sequences. The parallel synteny analysis was performed on *E*. *canis* YZ-1 from this study with the *Ehrlichia* reference sequences from NCBI, including *E. chaffeensis* (GenBank accession number CP000236) (**A**), *Ehrlichia* spp. (NZ_CP007474) (**B**), *E. muris* (CP006917) (**C**), *E. ruminantium* (CR767821) (**D**), *E. mineirensis* (NZ_CDGH01000070) (**E**), and *E. canis* (NC_007354) (**F**). The genomic differences/similarities in collinearity, inversion, translocation, and translocation + inversion are shown in height and color, as indicated at the top of the figure.

**Figure 4 microorganisms-12-00125-f004:**
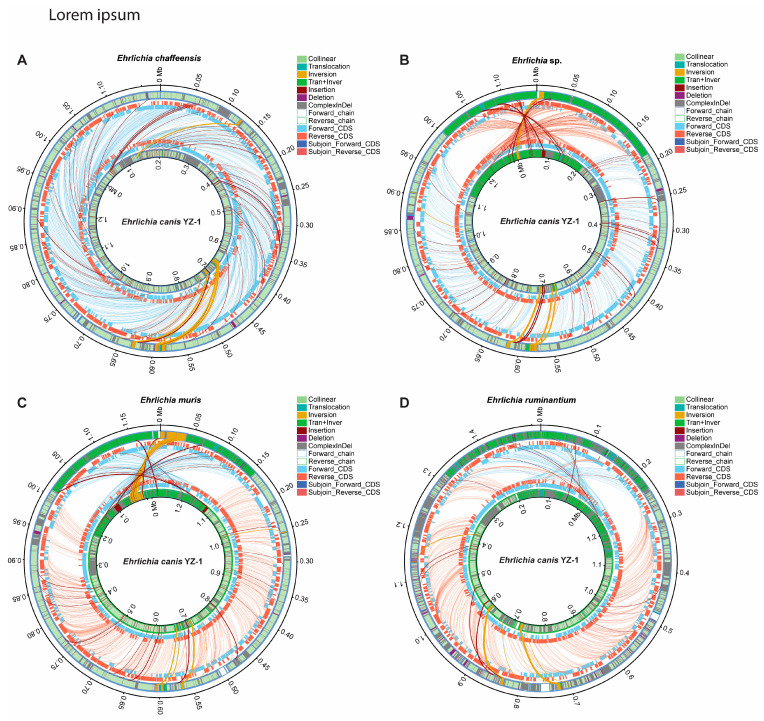
Comparative mutations analysis of *Ehrlichia canis* YZ-1 with the *Ehrlichia* reference sequences. The potential structural mutations were analyzed by arranging *E*. *canis* YZ-1 from this study with *Ehrlichia* reference sequences published before, including *E. chaffeensis* (CP000236) (**A**), *Ehrlichia* spp. (NZ_CP007474) (**B**), *E. muris* (CP006917) (**C**), *E. ruminantium* (CR767821) (**D**), *E. mineirensis* (NZ_CDGH01000070) (**E**), and *E. canis* (NC_007354) (**F**). The mutation types identified here were indicated in the figure accordingly.

**Table 1 microorganisms-12-00125-t001:** Basic information on the *Ehrlichia* strains used for analysis in this study.

Organism Name	GenBank #	No. of Genes	%G + C	No. of Bases	No. of Coding Bases	Coding Bases/Total Bases
*Ehrlichia canis* YZ-1	CP025749	1022	29.00	1,314,789	956,238	72.73
*Ehrlichia canis* str. Jake	NC_007354	985	28.96	1,315,030	959,246	72.94
*Ehrlichia chaffeensis* str. Arkansas	NC_007799	1158	30.10	1,176,248	945,019	80.34
*Ehrlichia* spp. HF	NZ_CP007474	988	29.65	1,148,904	879,236	76.53
*Ehrlichia muris* AS145	NC_023063	964	29.66	1,196,717	904,005	75.54
*Ehrlichia ruminantium* Welgevonden	NC_005295	976	27.48	1,512,977	959,837	63.44
*Ehrlichia minasensis* strain UFMG-EV	NZ_CDGH01000070	1119	29.89	1,366,818	982,241	71.86

Data obtained from IMG (v1.1; http://img.jgi.doe.gov accessed on 1 December 2023).

**Table 2 microorganisms-12-00125-t002:** Nuclear repeats analysis on the *Ehrlichia canis* YZ-1.

Repeat Type	Number	Total Length (bp)	Percentage in the Genome (%)
**Interspersed repeats**			
Long-terminal repeats	34	2405	0.1829
DNA transposons	14	1339	0.1018
Long interspersed nuclear elements	8	808	0.0615
Short interspersed nuclear elements	4	225	0.0171
Rolling circle	3	288	0.0219
Unknown	2	152	0.0116
**Tandem repeats**			
Tandem repeats	105	33,841	2.5739
Minisatellite DNA	52	2500	0.1901
Microsatellite DNA	1	28	0.0021

## Data Availability

The *E. canis* strain YZ-1 whole genome sequence has been deposited in GenBank under the accession number CP025749.

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
