# Peer review of "Whole Genome Sequencing and Comparative Analysis of the First Ehrlichia canis Isolate in China"

_microorganisms, 2024, doi:10.3390/microorganisms12010125_

Round 1

Reviewer 1 Report

Comments and Suggestions for Authors

This manuscript is well-structured and well-written.

The manuscript "Whole genome sequencing and comparative analysis of the first Ehrlichia canis isolate in China" is interesting and explore/opens avenues for innovative medications, development of new detection markers, and progress in vaccine development for ehrlichiosis. The topic is original. According to the literature, genomic information about this pathogen remains limited in China. Additionally, Ehrlichia canis is one of the most important species worldwide. The paper is well written and the conclusions are appropriate. However, it is necessary further studies about divergence or characterization of genes to conduct therapeutic interventions (vaccines) in the ongoing battle against tick-borne pathogens.

Author Response

Dear Reviewer,

Happy New Year! Please find the responses to your comments in the attached PDF file.

Thank you!

Reviewer 2 Report

Comments and Suggestions for Authors

Dear Authors, thank you for submitting an interesting article entitled „Whole genome sequencing and comparative analysis of the first Ehrlichia canis isolate in China” because the transfer of information is always useful.

Below you can find some general comments, strengths of the study and specific comments.

General comments:

I have carefully reviewed the manuscript titled " Whole Genome Sequencing and Comparative Analysis of the 2 First Ehrlichia canis Isolate in China" submitted to Microorganisms.

The study provides valuable insights into the genomic characteristics and evolutionary dynamics of Ehrlichia canis, a significant tick-borne pathogen causing canine monocytic ehrlichiosis (CME).

The paper aims to present a comprehensive genomic analysis of E. canis YZ-1, isolated from an infected dog in China. The complete genome sequencing revealed key features, including a genome size of 1,314,789 bp, 1,022 genes, 29% GC content, and 73% coding bases. Comparative analysis with other Ehrlichia species highlights conserved genes, suggesting evolutionary connections, particularly with E. ruminantium. The study emphasizes the intricate balance between pathogenicity and host adaptation, with a reduction in virulence-associated genes and the presence of a type IV secretion system (T4SS). The close relationship with E. canis Jake and E. chaffeensis, alongside variations with E. ruminantium and E. mineirensis, emphasizes the need for ongoing exploration of emerging strains and sequencing technologies.

The identification of conserved genes and potential evolutionary connections with E. ruminantium is a significant contribution.

The manuscript is generally clear and relevant, with well-organized sections.

Experimental design appears appropriate for the stated objectives.

Strengths:

One of the notable strengths of this research lies in its thorough characterization of E. canis YZ-1's genomic elements and functions. The inclusion of comparative analysis with representative genomes of other Ehrlichia species offers a broader perspective on the evolutionary connections and highlights conserved genes, particularly with E. ruminantium. The paper successfully navigates the complex landscape of Ehrlichia genomics, shedding light on potential evolutionary relationships that contribute to the overall understanding of this pathogen.

The identification of a type IV secretion system (T4SS) and the observed reduction in virulence-associated genes in E. canis YZ-1 add a layer of depth to the study, revealing an intricate balance between pathogenicity and host adaptation. The findings suggest potential avenues for innovative medications and the development of new detection markers, crucial for advancing our ability to combat ehrlichiosis.

The paper's emphasis on the close relationship with E. canis Jake and E. chaffeensis, coupled with nuanced genomic variations with E. ruminantium and E. mineirensis, underscores the importance of continuously exploring emerging strains and advancements in sequencing technologies. This forward-looking approach is commendable, as it not only contributes to our current knowledge but also sets the stage for future research directions and potential breakthroughs.

The insights provided in this study have significant implications for the field, particularly in the realms of probiotic resistance, vaccine development, and understanding host-pathogen interactions. By opening avenues for further investigations into the functional significance of identified genes, the paper paves the way for a more holistic comprehension of Ehrlichia's biology and its broader implications for pathogenicity and transmission.

In conclusion, this paper stands out as a valuable contribution to the scientific community, offering a rich source of genomic information on E. canis YZ-1 and providing a solid foundation for future research endeavors in the field of tick-borne pathogens and ehrlichiosis.

Specific comments:

Lines: 18, 125, 145, 146, 226, 240, 244, 246 and throughout the text Please change Ehrlichia sp. with Ehrlichia spp. Actually, sp. becomes spp. for all species.

The legend in Figure 1 is impossible to read. Could you please review Figure 3E as well?

Author Response

(The authors gave the same response as above.)

Reviewer 3 Report

Comments and Suggestions for Authors

This article presents novel information about the genetics of Ehrlichia canis.  Please try to avoid repetition of information (e.g. lines 19-20). The tables presented in the manuscript should be attached in a supplementary file to facilitate their reading. Some minor typos should be modified (e.g. line 34)

Comments on the Quality of English Language

I suggest minor editing of the language

Author Response

(The authors gave the same response as above.)
